# Official Websites of Local Health Centers in Taiwan: A Nationwide Study

**DOI:** 10.3390/ijerph16030399

**Published:** 2019-01-31

**Authors:** Ya-Chuan Hsu, Tzeng-Ji Chen, Feng-Yuan Chu, Hao-Yen Liu, Li-Fang Chou, Shinn-Jang Hwang

**Affiliations:** 1Department of Family Medicine, Taipei Veterans General Hospital, No. 201, Sec. 2, Shi-Pai Road, Taipei 112, Taiwan; ych97160@gmail.com (Y.-C.H.); steven2259898@gmail.com (F.-Y.C.); yen.ee93@gmail.com (H.-Y.L.); sjhwang@vghtpe.gov.tw (S.-J.H.); 2School of Medicine, National Yang-Ming University, No. 155, Sec. 2, Linong Street, Taipei 112, Taiwan; 3Department of Public Finance, National Chengchi University, Taipei 116, Taiwan; lifang@nccu.edu.tw

**Keywords:** public health administration, Internet, patient portals, Taiwan

## Abstract

Local health centers (LHCs) play a key role in public health. Because it has now become popular to seek health information on the Internet, an effective website is indispensable to an LHC. Our study aimed to survey the official websites of LHCs in Taiwan with an evaluation framework. All 369 LHCs in Taiwan were surveyed in March 2018. The evaluation indicators included health information, online interactive services, technical features, institutional information, links to external resources, website management, the last updated time, and number of visitors. The indicators were stratified by the urbanization levels of the LHCs. In total, 98.0% (*n* = 360) of the LHCs had official websites. The majority (*n* = 241) of the websites were updated within the past 30 days, and most of the websites (*n* = 353) provided health information. However, the information provided varied considerably. Few LHCs (*n* = 31) provided online interactive services in terms of an online appointment function. In terms of providing online consultation services, rural LHCs outperformed suburban and urban LHCs (16.4% versus 14.5% and 6.0%, respectively). Most LHCs in Taiwan do not seem to take full advantage of the Internet, with their websites typically serving as static bulletin boards instead of new channels of communication. Further studies could focus on the effectiveness of these websites.

## 1. Introduction

Local health centers (LHCs) are the cornerstone of public health in Taiwan and can be traced back to the Japanese colonial era (1895–1945) [1]. Each town or district of a city has an LHC that handles a variety of responsibilities, including the provision of medical care, disease surveillance, and health promotion [2]. In Taiwan, nearly all residents can frequently and easily seek medical services among the various types of healthcare facilities under the universal coverage of the National Health Insurance (NHI) program since 1995 [3]. Private clinics and outpatient departments of hospitals play their roles in primary healthcare, and the LHCs mainly deal with public health issues in urban and suburban areas. However, the LHCs cover the majority of primary care services in some rural areas [4]. The traditional method of disseminating health information via posters and pamphlets has been giving way, in recent years, to dissemination via digital media. Relatedly, previous studies have reported the huge popularity of using the Internet for multiple health-related purposes [5,6]. An individual’s decision-making behavior thus might be potentially influenced by his or her online health searches [7]. In addition, the Internet is not only a powerful gateway to health information but also a medium that can be used to strengthen social interactions between health information seekers and providers [8,9]. More importantly, the increasing ubiquity of Internet access narrows certain disparities between rural and urban areas. In December 2017, the Internet penetration rate in Taiwan had reached 87.9%, ranking third in Asia after the rates in Japan (93.9%) and South Korea (92.6%) [10,11]. Nowadays, an effective and reliable website is thus indispensable to an LHC. This study aimed to conduct a nationwide survey of all LHC websites in Taiwan and compare the websites for LHCs with different levels of urbanization. The results of the study will enhance the understanding of Internet use by LHCs and provide useful recommendations to assist local healthcare providers in improving medical outcomes by reinforcing the functionality of the LHC websites.

## 2. Materials and Methods

### 2.1. Data Collection

This descriptive study was performed in February and March of 2018. There were 369 LHCs in Taiwan as of March 2018. They were administered by 22 health bureaus of counties/cities, which were all in turn supervised by the Health Promotion Administration of the Ministry of Health and Welfare in Taiwan.

Taiwan comprises 6 special metropolitan cities (Taipei, New Taipei, Taoyuan, Taichung, Tainan, and Kaohsiung), 13 counties (Hsinchu, Miaoli, Changhua, Nantou, Yunlin, Chiayi, Pingtung, Yilan, Hualien, Taitung, Penghu, Kinmen, and Lienchiang), and 3 cities (Keelung, Hsinchu, and Chiayi). We searched the website of each health bureau and then identified the official websites of its subordinate LHCs. We also cross-checked the website of each LHC by conducting online searches for the name of the given LHC to see whether the public could find the website directly through search engines.

The locations of the LHCs were categorized according to the urbanization stratification of Taiwan’s 369 townships developed by Taiwan’s National Health Research Institutes [12]. Of the seven urbanization levels in that stratification, we defined levels 1 and 2 as urban, levels 3 and 4 as suburban, and levels 5 to 7 plus the remote islands as rural (Table 1).

### 2.2. Features of LHC Websites

We first examined the basic features and contents of all of the LHC websites and summarily categorized 18 items under 6 dimensions.

#### 2.2.1. Health Information

LHCs are charged with informing, educating, and empowering people with regard to health issues [13]. This study investigated the presence of health promotion materials on each website, e.g., health education materials, disease surveillance materials, and the latest breaking news on epidemics or pandemics. Any posts regarding upcoming events in the LHC were considered health promotion materials as well.

#### 2.2.2. Online Interactive Functions

According to a previous study, a virtual communication is effective when an online user feels that it has the familiarity of a face-to-face interaction [14]. For this dimension of online interactive functions, this study surveyed the LHC websites for the presence of real-time communication between healthcare seekers and providers, which included online consultations, online appointment functions, and mechanisms for providing feedback to the given LHC. An online appointment can enhance a patient’s willingness to seek medical care by reducing time and location restrictions [15]. With respect to feedback mechanisms, this study checked each website to see if it offered a virtual space in which individual users could voice their opinions.

#### 2.2.3. Institutional Information

Because the frequency with which people conduct online searches for miscellaneous health-related subjects has been growing rapidly, it is vital for an LHC to provide explicit and valid institutional information on its website [16]. In terms of this dimension, then, the current study checked each website’s capability to provide adequate information about the LHC itself. We focused on the following items that are most commonly sought by patients—contact details, doctor profiles, and lists of services.

#### 2.2.4. Technical Features

This dimension can reflect a website’s functionality, including its transparency, accessibility, and efficiency [17]. We investigated whether each website had an internal search engine and a site map that allowed users to navigate through the site easily to find the information they were looking for. In addition, the study also surveyed the availability of website features in different languages, which could increase the accessibility of a site by enabling larger populations to acquire health information.

#### 2.2.5. External Activities

A previous study reported that linking people to necessary personal medical services and social support was one of the key indicators for assessing the quality of healthcare provided at all levels of health systems [13]. Thus, we investigated whether each LHC website provided links to health authorities and useful community resources. Furthermore, some studies have suggested that social media is an emerging new tool to improve public health by allowing evidence-based information to be spread efficiently [18,19]. As such, links to different social media sites and apps were also included in this dimension.

#### 2.2.6. Management and Utilization

Generally, this dimension was used to assess the management and utilization of a website. More specifically, we inspected the number of visitors to each website to evaluate its utilization. We grouped the number of visitors reported on each website into six categories (2,500,000–3,000,000; 2,000,000–2,500,000; 1,500,000–2,000,000; 1,000,000–1,500,000; 50,000–1,000,000; and <50,000), and then compared the results for the LHC with different urbanization levels. We took the last updated time as an indicator of website management. We categorized the updated times into three groups (within 30 days, between 31 and 90 days, and more than 90 days ago). Due to a lack of updated time references on some websites, we used the date of the latest post on the given website as a proxy for the updated times for those sites.

### 2.3. Statistical Analysis

Descriptive statistics were calculated. Additionally, box-and-whisker plots were constructed to display the distribution of the number of visitors for the LHC websites according to urbanization levels.

## 3. Results

### 3.1. Distribution of the LHCs

Of the 369 LHCs in Taiwan as of March 2018, the highest proportions were located in rural (42%, 155/369) and suburban (39.3%, 145/369) areas. Furthermore, 134 LHCs were located in Southern Taiwan, 108 were located in Central Taiwan, 89 were located in Northern Taiwan, 29 were located in Eastern Taiwan, and 9 were located in remote islands. Only the 9 LHCs located in remote islands did not have an official website in operation. Those LHCs were located in Kinmen County and Lienchiang County, which were stratified as rural-level counties (Table 1).

### 3.2. Features of the LHC Websites

The distributions of the characteristics of the LHC websites across different urbanization levels are listed in Table 2. As for health information, almost all of the LHCs (98.1%, 353/360) posted health education articles on their websites. However, the quantity of posts and the quality of their contents varied greatly. For example, some of the posted health information was out of date. Significantly fewer rural LHCs than suburban and urban LHCs posted recent medical news (21.9% versus 47.6% and 64.0%, respectively). Regardless of the urbanization levels of the LHCs, most of the websites could be fully utilized as bulletin boards for posting upcoming events.

Regarding online interactive functions, the rural LHCs outperformed the suburban and urban LHCs in terms of providing online consultations (16.4% versus 14.5% and 6.0%, respectively). Nonetheless, the general performance in this dimension was unsatisfactory for all urbanization levels. As for the LHC websites which provided online consultation services, all of them were presented in the form of communication through e-mails or message boards. Very few websites provided an online appointment function (8.6%, 31/360), and there was nearly a complete absence of online appointment functions on the websites of the rural LHCs (4.8%, 7/146). Meanwhile, 85.6% of the websites had some mechanism for providing feedback via a message board, e-mail, or Internet forum. However, we found that some of the open forums were inactive or unregulated.

In terms of institutional information, all of the websites provided the contact details of the postal address and telephone number of the corresponding LHC. However, communication via e-mail was supported by only about half of the websites (57.2%, 206/360). Meanwhile, although most of the websites listed the content of services (94.7%), less than half (47.5%) provided doctor profiles. In general, rural LHCs had relatively insufficient institutional information.

With regard to technical features, the accessibility tools were generally adequate. The vast majority of the websites had a sitemap (95.6%) and an internal search engine (94.4%). Unfortunately, less than a third of the websites (29.2%) had a second language version (i.e., English). There was an absolute lack of versions of Southeast Asian languages. In addition, for those websites which did have an English version, we found that only a brief description of the given LHC’s history was typically presented in English.

As for external activities, all of the websites were linked from their supervising administrative agencies and all of them provided links to sites with useful resources (e.g., general health information, other healthcare institutions, and educational institutions). It is worth mentioning that only 20% of the LHCs had accounts on social media platforms (i.e., Facebook, Instagram, and Line), but all of them had a link or QR code (Quick Response code) marker on the website linking to its adopted social media.

With regard to management and utilization, 76.1% of the websites contained a statement of the given LHC’s privacy and security policy. Most of the LHC websites (93.6%) had also been updated within the last two months. A higher proportion (96%, 66/69) of urban LHCs had updated their websites during the last two months. Of all the LHC websites, 66.9% had their last updated time within 30 days, 22.8% had their last update between 31 and 90 days, and only 9.4% had their last update more than 90 days ago (Table 3). A higher proportion (74%, 51/69) of urban LHCs than suburban and rural LHCs (61.4% and 69.2%, respectively) had updated their websites within 30 days.

### 3.3. Number of Visitors Distribution

Of the 360 LHC websites, 251 had a visitor counter feature, including 48 of the sites for urban LHCs, 99 for suburban LHCs, and 104 for rural LHCs. The websites of LHCs in urban areas had a tendency to have higher number of visitors than the websites of those in suburban and rural areas (on average 1,677,640 versus 336,558 and 137,735, respectively). The websites of the LHCs in urban areas had a median of 659,371 visitors, with a first quartile of 146,927 visitors and a third quartile of 1,402,733.5 visitors. For the websites of LHCs in suburban areas, the median was 171,552 visitors, the first quartile was 42,408 visitors, and the third quartile was 411,807 visitors. The websites of LHCs in rural areas had a median of 80,101 visitors, a first quartile of 47,455 visitors, and a third quartile of 174,247 visitors (Figure 1).

## 4. Discussion

### 4.1. Principal Findings

The official websites have a great potential to boost communication between LHCs and citizens and optimize the dissemination of health information universally. Despite the fact that some studies evaluating LHC websites have been conducted in other countries, the quantity of such studies is still small [20,21,22,23,24,25,26]. Our findings showed that 98% (360/369) of the LHCs surveyed in this study had an official website as of March 2018. The urbanization level of the LHCs was not related to whether or not they had an official website. Overall, the frequency with which the websites were updated was insufficient. Though all websites contained health information, some of them were inaccurate, incomplete, or out of date.

### 4.2. Features of LHC Websites

Visitors seek a diverse range of information pertaining not only to treatment options, which are typically their primary concern, but also to the healthcare institution itself. LHCs should thus provide citizens with accurate and validated information for decision-making purposes. In our study, we found that the quality of LHC websites varied widely. Most websites shared similar website architectures, but a small portion of websites were unregulated. Few websites had interactive functions, especially in terms of online appointment and online consultation functions. An online appointment function is helpful because it allows patients to make an appointment with a doctor of their choice with reduced obstacles of time and location (i.e., a patient does not have to travel to the LHC to make the appointment in person). For healthcare providers, meanwhile, an online appointment function makes them aware of a given patient’s situation and gives them adequate time to prepare the necessary information before meeting the patient [15]. The almost complete lack of online appointment functions for the websites of rural LHCs could be a reflection of the lower Internet penetration rates in rural areas [27]. Otherwise, while most of the LHC websites had a feedback mechanism, most of those mechanisms were somewhat unregulated. Compared to most commercial websites, which typically seek customer opinions actively to improve business performance, the LHC websites did not seem to pay much attention to the feedback from patients or website users. Further research is required to investigate the attitudes and thoughts of LHC personnel toward the value of feedback mechanisms, as well as the influence of such feedback on medical practice. A useful website is a live website that is visited regularly by users and provides constant updates of its content and interactions with users in order to satisfy their needs [28]. LHCs should thus measure the benefits brought by adding online appointment functions and strengthening their online consultation functions.

We found that most of the websites provided adequate contact information to facilitate direct and interpersonal contact with the LHC, but said contact information seemed to mainly focus on traditional communication channels, e.g., by consisting of phone numbers and postal addresses, instead of digital channels. These facts might reflect citizens’ preferences for verbal communication. With respect to technical features, however, good website accessibility is vital for visitors to utilize a website to the greatest extent possible. The findings of this study indicated that most of the websites were easy to navigate. However, very few of the websites had second language versions. Nonetheless, according to the statistics from the National Immigration Agency, new immigrants to Taiwan recently accounted for 3% of the total population and that percentage is still increasing. Over ninety percent of immigrants were from Southeast Asian countries [29]. The lack of multilingual functions can thus be considered a significant limitation for web accessibility. LHC websites should thus add versions in other languages in order to address the needs of immigrant populations.

Few of the websites for LHCs in rural areas provided doctors’ profiles, which might indicate that the LHCs did not need to provide doctor introductions for advertising purposes to attract patients because the LHCs were the main or even the only healthcare provider in those rural areas [30].

In Taiwan, 51.1% of the hospitals have been found to have an official Facebook fan page. Even among local community hospitals, 46.1% had an official Facebook fan page [31]. In contrast, this study found that only 20.0% of the LHCs in Taiwan had social media accounts (e.g., Facebook, Instagram, or Line). Social media have a great potential for health information dissemination across the public health system [18,19]. Therefore, social media adoption should be considered for LHCs in order to educate the general public about accurate health information more efficiently. In general, however, the LHCs seemed to pay more attention to website development than maintenance. For better website functionality, we suggest that skilled IT professionals be hired to manage the LHC websites in Taiwan.

### 4.3. Distribution of “Visitor Numbers”

The number of visitors to a website can serve as one of the web analytic metrics for a website’s utilization and conversion rate. The utilization and conversion rate have a positive correlation with the function of a website [28]. Our findings showed that the websites of rural and suburban LHCs had lower numbers of visitors than those of urban LHCs. However, the statistics from Taiwan’s Ministry of Health and Welfare reported that the preventive service volumes of suburban and rural LHCs were generally higher than those of urban LHCs in 2016 [32]. The number of website visitors was thus disproportionate to the volume of preventive services of the LHCs of different urbanization levels. Being the major or even the sole healthcare provider in some rural areas, the LHCs didn’t earn more website visitors than those in suburban and urban areas. The remarkably lower utilization of websites operated by LHCs in rural and suburban areas might be attributed to rural–urban disparities in terms of population density and digitalization [27,33].

### 4.4. Limitations

There are several limitations to this study. First, the Internet is constantly changing. We could thus only take a snapshot of each site’s dynamic performance at a particular time (e.g., the last updated time and the number of visitors). However, these indicators might be influenced by particular events at various times, e.g., the number of website visitors might increase in flu seasons because citizens might actively look for the release date of government-funded vaccination. Thus, the results in one study may not be correlated with those of another similar study at different times. Second, all of the LHCs subordinate to the same Public Health Bureau shared similar website architectures. The Public Health Bureau in every county/city had its own website developer who made a website template for those LHCs administered by that bureau. This means that the disparities in website quality could be partially attributed to technology resources, not only to urbanization levels. Third, we evaluated the number of visitors by checking the visitor counters presented on the homepage of each website. However, approximately one-third (30.2%, 109/360) of the websites did not have this function. Furthermore, the number of visitors indicated by such counters is cumulative. As such, the actual number of website visitors might be inflated by repeat visits by the same people. In addition, older websites would have had more time to accumulate a larger number of website visitors. Fourth, we could only quantify the number of posts updated in the websites, but the actual quality of posts needs to be investigated further. We also could not determine whether the number of visitors was related to the quality of information provided by the LHC or the level of interaction between users and the LHC. Future studies are thus required to provide a multidimensional measurement of website quality.

## 5. Conclusions

The LHCs play a key role in providing citizens with medical services in most suburban and rural areas. The websites of LHCs could thus potentially be a viable force for successful public health programs, but most of the LHCs in Taiwan do not seem to take full advantage of the Internet. It is thus worth making an effort to reinforce website management for all LHCs in Taiwan. Meanwhile, further study is necessary to evaluate the benefits to public health outcomes contributed by these websites.

## Figures and Tables

**Figure 1 ijerph-16-00399-f001:**
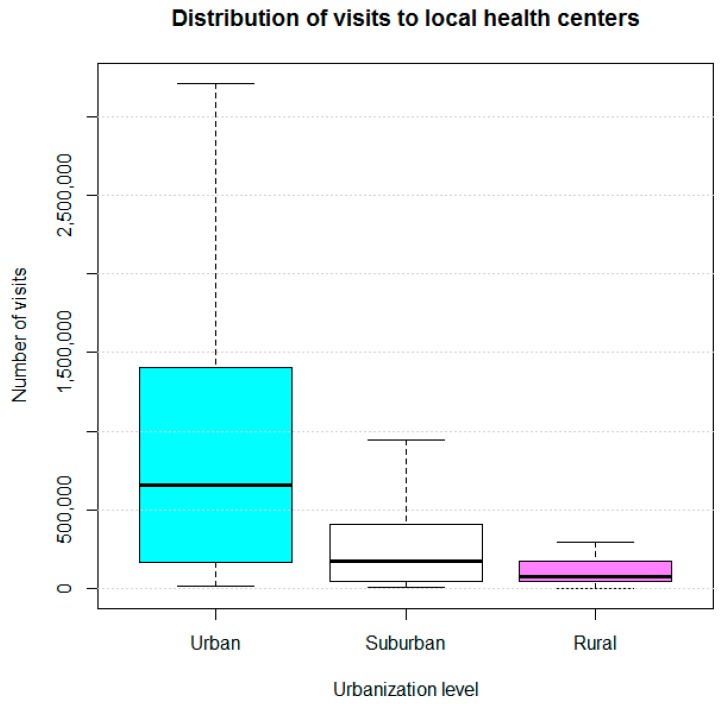
Number-of-visitors distribution of websites of 251 LHCs in Taiwan.

**Table 1 ijerph-16-00399-t001:** Distributions of local health centers (LHCs) in Taiwan (*n* = 369).

Cities and Counties	Urbanization Levels			Total
	Urban	Suburban	Rural	
Northern Taiwan				
Keelung City	4	3	0	7
Taipei City	12	0	0	12
New Taipei City	10	16	3	29
Taoyuan City	4	8	1	13
Hsinchu City	2	1	0	3
Hsinchu County	2	8	3	13
Yilan County	2	7	3	12
Central Taiwan				
Taichung City	9	17	4	30
Miaoli County	1	10	7	18
Changhua County	2	15	10	27
Nantou County	0	3	10	13
Yunlin County	1	5	14	20
Southern Taiwan				
Tainan City	5	18	14	37
Kaohsiung City	11	17	9	37
Chiayi City	2	0	0	2
Chiayi County	0	5	13	18
Pingtung County	1	5	27	33
Penghu County	0	3	4	7
Eastern Taiwan				
Hualien County	1	3	9	13
Taitung County	0	1	15	16
Remote islands				
Kinmen County	0	0	5	5
Lienchiang County	0	0	4	4
Total	69	145	155	369

**Table 2 ijerph-16-00399-t002:** Features of official websites of all LHCs (*n* = 360).

Features	Urban *n* = 69, (%)	Suburban *n* = 145, (%)	Rural *n* = 146, (%)	Total *n* = 360, (%)
Health information				
Health education articles	69 (100)	143 (98.6)	141 (96.6)	353 (98.1)
Recent medical news	44 (64)	69 (47.6)	32 (21.9)	145 (40.3)
Upcoming events	68 (99)	138 (95.2)	135 (92.5)	341 (94.7)
Interactive online services				
Online appointment	7 (10)	17 (11.7)	7 (4.8)	31 (8.6)
Online consultation	4 (6)	21 (14.5)	24 (16.4)	49 (13.6)
Feedback mechanism	55 (80)	133 (91.7)	120 (82.2)	308 (85.6)
Institutional information				
Contact details: postal address, telephone number	69 (100)	145 (100.0)	146 (100.0)	360 (100.0)
Contact details: e-mail	43 (62)	89 (61.4)	74 (50.7)	206 (57.2)
Doctor profiles	35 (51)	85 (58.6)	51 (34.9)	171 (47.5)
List of services	67 (97)	142 (97.9)	132 (90.4)	341 (94.7)
Technical items				
Sitemap	68 (99)	142 (97.9)	134 (91.8)	344 (95.6)
Website searcher	68 (99)	139 (95.9)	133 (91.1)	340 (94.4)
Multiple language	23 (33)	43 (29.7)	39 (26.7)	105 (29.2)
External activities				
Links from related authorities	69 (100)	145 (100.0)	146 (100.0)	360 (100.0)
Links to related community resources	69 (100)	145 (100.0)	146 (100.0)	360 (100.0)
Links to adopted social media (i.e., Facebook, Instagram, Line)	16 (23)	29 (20.0)	27 (18.5)	72 (20.0)
Management and utilization				
Privacy and security policy	51 (74)	118 (81.37)	105 (71.9)	274 (76.1)
Web page updated during the last two months	66 (96)	139 (95.86)	132 (90.4)	337 (93.6)

**Table 3 ijerph-16-00399-t003:** Last updated time on websites of 360 LHCs in Taiwan.

Last Updated Time	Urban *n* = 69 (%)	Suburban *n* = 145 (%)	Rural *n* = 146 (%)	Total *n* = 360 (%)
Within 30 days	51 (74)	89 (61.4)	101 (69.2)	241 (66.9)
Between 31 and 90 days	13 (19)	37 (25.5)	32 (21.9)	82 (22.8)
More than 90 days	5 (7)	19 (13.1)	13 (8.9)	37 (10.3)

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
