# Peer review of "Official Websites of Local Health Centers in Taiwan: A Nationwide Study"

_ijerph, 2019, doi:10.3390/ijerph16030399_

Round 1
Reviewer 1 Report
This manuscript presents findings of a survey of official websites of all local health centres in Taiwan. The paper is well structured and clearly written. The data presented in the paper should be useful for monitoring the development of web-based communications for local health centres over time, and will facilitate comparisons between different types of health care facilities within the country and with other countries. I only have a small number of comments which I hope will help improve the manuscript. The comments mainly relate to the precision in the use of terms and provision of information that would be interesting/useful for international readers.
(1) Throughout the manuscript the authors have used the term “health education” to refer to health-related information provided on the official websites of the local health centres. As health education normally involves more than just passing on information to patients, use of the term “health information” instead of “health education” is probably more accurate and appropriate (e.g. the sub-heading for section 2.2.1 and in many other places in the article).
(2) Line 110 subsection heading for section 2.2.6: what is covered in this section is better described as “utilization” or “usage” rather than “usability” (which refers to how easy it is for users to find the information that they want or to carry out the action that they intended).
(3) Line 111 – following from the comment above, it might be better to describe this dimension as being related to assessment of “management and utilization” of the web site rather than “the overall quality”, which is a different concept as the authors highlighted in Section 4.4.
(4) Online consultation: this is a feature that many health systems across the world is trying to develop, and therefore is likely to be of great interest to readers. Would it be possible to provide some more information on this in the Results section, e.g. in what format is the online consultation offered - online live chat, videoconference call, communication through e-mails or something else; or if the information is not available, the authors may wish include some discussion (e.g. acceptability, feasibility, usage) surrounding this in the Discussion section?
(5) Background: for the benefit of international readers to understand the context, the authors may wish to describe the pattern of primary health care provision and utilization in a bit more detail in the Introduction section, for example (please note the following texts are based on my understanding which isn’t necessarily correct; it would be good [where possible] if the authors could support such descriptions with some citations/data): in urban and suburban settings, public and private hospital outpatient departments and private clinics cover the majority of primary medical care services, and public funded local health centres mainly deal with disease surveillance and health promotion functions, whereas in rural settings the local health centres are sometimes the only health services that are available. It might also be useful for readers to know that irrespective of service providers, the vast majority of health services are covered by Taiwan’s National Health Insurance scheme.
(6) “doctor’s profiles” might be a better term to use than “doctor introductions”.
(7) Lines 157-162: given that there is a growing population of foreign workers and immigrants from various countries in South East Asia who do not necessarily have good comprehension of English, it would be useful to know if any of the websites include any additional language other than English.
(8) Lines 264-265: this study essentially covers a complete national sample, and therefore the finding should be robust. You might wish to emphasise this strength, and maybe also to clarify that the results in one study may not be correlated with those of another similar study “conducted at different time or in different countries”.
Author Response
Dear Sir,
Thank you very much for reviewing our manuscript entitled "Official Websites of Local Health Centers in Taiwan: A Nationwide Study" submitted to International Journal of Environmental Research and Public Health.
We would like to thank the article reviewers for their valuable comments and we have made point-to-point responses to the reviewers' comments.
Please refer to the attachment (ResponsesToReviewer1_IJERPH_20190125R).
Sincerely yours,
Ya-Chuan Hsu, MD
Resident Doctor
Department of Family Medicine, Taipei Veterans General Hospital, Taipei, Taiwan
Feng-Yuan Chu, MD
Resident Doctor
Department of Family Medicine, Taipei Veterans General Hospital, Taipei, Taiwan
Hao-Yen Liu, Dr. med.
Department of Family Medicine, Taipei Veterans General Hospital, Taipei, Taiwan
Tzeng-Ji Chen, Dr. med.
Professor
Institute of Hospital and Health Care Administration, School of Medicine, National Yang-Ming University, Taipei, Taiwan
Director
Department of Family Medicine, Taipei Veterans General Hospital, Taipei, Taiwan
Li-Fang Chou, M.Sc. (Public Health), Dr. sc. pol.
Professor
Department of Public Finance, National Chengchi University, Taipei, Taiwan
Shinn-Jang Hwang, MD, FACG.
Vice Superintendent
Taipei Veterans General Hospital, Taipei, Taiwan
Professor
Department of Family Medicine, School of Medicine, National Yang-Ming University, Taipei, Taiwan

Reviewer 2 Report
our study is important toward improving health care access for the Taiwanese people. I appreciate your writing style and generally clear presentation. Your report of methods is transparent. A majority of comments offered below are minor and refer to potential/required edits of writing.
Line 48: Delete 'of the'
Line 70: Recommend changing to: '...first examined basic features...'
Line 71: Recommend changing to: 'summarily categorized 18...'
Line 80: Change to: 'feels'
Line 99: Change to: '...availability of website...'
Lines 113-114: It is difficult to easily grasp the value of the larger numbers without the use of commas. I suggest, for example, that you rewrite 2,500,000-3,000,000. Additionally, why do you drop from 1,000,000 to <50,000 from category 4 to 5 when all other transitions are natural/logical next values?
Line 168: Change to: '...contained a statement...'
Lines 183-188: See comment #6 regarding using commas to differentiate the digits/values of large numbers.
Figure 1: Y axis: See comments #6, 8
Line 193: Suggest changing 'huge' to 'great'
Line 233: Change to: 'immigrants'
Line 244: Recommend deleting the two uses of 'on'
Line 245: Recommend removing comma after 'Therefore'
Line 249: Recommend deleting 'must'
Lines 257-258: The sentence should be rewritten as it is unclear.
Line 258: The word 'function' as used in this context is unclear. Please use a more fitting term.
References: For the following references please capitalize the first word after a colon (:) in the title: #3, 15, 18, 20, 23.
Reference 29: Change to 'hospitals' in title.
Author Response
Dear Sir,
Thank you very much for reviewing our manuscript entitled "Official Websites of Local Health Centers in Taiwan: A Nationwide Study" submitted to International Journal of Environmental Research and Public Health.
We would like to thank the article reviewers for their valuable comments and we have made point-to-point responses to the reviewers' comments.
Please refer to the attachment (ResponsesToReviewer2_IJERPH_20190125R).
Sincerely yours,
Ya-Chuan Hsu, MD
Resident Doctor
Department of Family Medicine, Taipei Veterans General Hospital, Taipei, Taiwan
Feng-Yuan Chu, MD
Resident Doctor
Department of Family Medicine, Taipei Veterans General Hospital, Taipei, Taiwan
Hao-Yen Liu, Dr. med.
Department of Family Medicine, Taipei Veterans General Hospital, Taipei, Taiwan
Tzeng-Ji Chen, Dr. med.
Professor
Institute of Hospital and Health Care Administration, School of Medicine, National Yang-Ming University, Taipei, Taiwan
Director
Department of Family Medicine, Taipei Veterans General Hospital, Taipei, Taiwan
Li-Fang Chou, M.Sc. (Public Health), Dr. sc. pol.
Professor
Department of Public Finance, National Chengchi University, Taipei, Taiwan
Shinn-Jang Hwang, MD, FACG.
Vice Superintendent
Taipei Veterans General Hospital, Taipei, Taiwan
Professor
Department of Family Medicine, School of Medicine, National Yang-Ming University, Taipei, Taiwan

Reviewer 3 Report
- Authors have done a good experimentation but the problem of this paper is more technical paper than research paper.
- I suggest ti improve it by designing a model to help customers to identify efficiently the information and LHC.
- Also, when the authors say :
The results of the study will enhance 50 the understanding of Internet use by LHCs and provide related guidance to assist local 51 healthcare providers in facilitating medical services.
I did not find the guidance to facilitate medicsl services ?
Author Response
Dear Sir,
Thank you very much for reviewing our manuscript entitled "Official Websites of Local Health Centers in Taiwan: A Nationwide Study" submitted to International Journal of Environmental Research and Public Health.
We would like to thank the article reviewers for their valuable comments and we have made point-to-point responses to the reviewers' comments.
Please refer to the attachment (ResponsesToReviewer3_IJERPH_20190125R).
Sincerely yours,
Ya-Chuan Hsu, MD
Resident Doctor
Department of Family Medicine, Taipei Veterans General Hospital, Taipei, Taiwan
Feng-Yuan Chu, MD
Resident Doctor
Department of Family Medicine, Taipei Veterans General Hospital, Taipei, Taiwan
Hao-Yen Liu, Dr. med.
Department of Family Medicine, Taipei Veterans General Hospital, Taipei, Taiwan
Tzeng-Ji Chen, Dr. med.
Professor
Institute of Hospital and Health Care Administration, School of Medicine, National Yang-Ming University, Taipei, Taiwan
Director
Department of Family Medicine, Taipei Veterans General Hospital, Taipei, Taiwan
Li-Fang Chou, M.Sc. (Public Health), Dr. sc. pol.
Professor
Department of Public Finance, National Chengchi University, Taipei, Taiwan
Shinn-Jang Hwang, MD, FACG.
Vice Superintendent
Taipei Veterans General Hospital, Taipei, Taiwan
Professor
Department of Family Medicine, School of Medicine, National Yang-Ming University, Taipei, Taiwan

Round 2
Reviewer 3 Report
This paper could be published.